# Relationship of the Frequency, Distribution, and Content of Meals/Snacks to Glycaemic Control in Gestational Diabetes: The myfood24 GDM Pilot Study

**DOI:** 10.3390/nu12010003

**Published:** 2019-12-18

**Authors:** Michelle A. Morris, Jayne Hutchinson, Carla Gianfrancesco, Nisreen A. Alwan, Michelle C. Carter, Eleanor M. Scott, Janet E. Cade

**Affiliations:** 1Leeds Institute for Data Analytics, School of Medicine, Level 11 Worsley Building, University of Leeds, Leeds LS2 9JT, UK; 2Nutritional Epidemiology Group, School of Food Science and Nutrition, University of Leeds, Leeds LS2 9JT, UK; j.hutchinson1@leeds.ac.uk (J.H.); carla.gianfrancesco@nhs.net (C.G.); michellecarter.uk@hotmail.co.uk (M.C.C.); j.e.cade@leeds.ac.uk (J.E.C.); 3Sheffield Diabetes and Endocrine Centre, Sheffield Teaching Hospitals NHS Trust, Sheffield S10 2JF, UK; 4School of Primary Care, Population Sciences and Medical Education, Faculty of Medicine, University of Southampton, Southampton SO16 6YD, UK; n.a.alwan@soton.ac.uk; 5NIHR Southampton Biomedical Research Centre, University of Southampton and University Hospital Southampton NHS Foundation Trust, Southampton SO16 6YD, UK; 6Leeds Institute for Cardiovascular and Metabolic Medicine, University of Leeds, Leeds LS2 9JT, UK; e.m.scott@leeds.ac.uk

**Keywords:** gestational diabetes mellitus, glycaemic control, online dietary assessment, snacking

## Abstract

This study examines nutritional intakes in Gestational diabetes mellitus piloting the myfood24 tool, to explore frequency of meals/snacks, and daily distribution of calories and carbohydrates in relation to glycaemic control. A total of 200 women aged 20–43 years were recruited into this prospective observational study between February 2015 and February 2016. Diet was assessed using myfood24, a novel online 24-h dietary recall tool. Out of 200 women 102 completed both ≥1 dietary recalls and all blood glucose measurements. Blood glucose was self-measured as part of usual care. Differences between groups meeting and exceeding glucose targets in relation to frequency of meal/snack consumption and nutrients were assessed using chi-squared and Mann–Whitney tests. Women achieving a fasting glucose target <5.3 mmol/L, compared to those exceeding it, consumed three meals (92% vs. 78%: *p* = 0.04) and three snacks (10% vs. 4%: *p* = 0.06) per day, compared with two or less; and in relation to evening snacks, consumed a higher percentage of daily energy (6% vs. 5%: *p* = 0.03) and carbohydrates (8% vs. 6%: *p* = 0.01). Achieving glycaemic control throughout the day was positively associated with snacking (*p* = 0.008). Achieving glucose targets was associated with having more snacks across the day, and may be associated with frequency and distribution of meals and nutrients. A larger study is required to confirm this.

## 1. Introduction

Gestational diabetes mellitus (GDM) is defined as carbohydrate intolerance with initial onset or first recognition in pregnancy [1]. The estimated prevalence is around 4% in England and Wales [2]. GDM usually resolves after giving birth, however, around 50% of cases develop type 2 diabetes (T2D) in their lifetimes [2]. The associated hyperglycaemia causes serious complications of pregnancy affecting both mother and baby. Thus, the main treatment aim is to restore blood glucose (BG) levels to normal throughout the remainder of the pregnancy [3].

Treatment initially focuses on diet and lifestyle changes with education and support provided by a dietitian. According to UK Guidelines, if within one to two weeks of dietary modification blood glucose (BG) levels are above the target range for pregnancy: fasting ≥5.3 mmol/L and 1-h postprandial ≥7.8 mmol/L, pharmacological treatment is initiated [2].

A recent Cochrane systematic review of 15 randomised controlled trials concluded that women with GDM receiving non-pharmacological lifestyle interventions including dietary modification with at least one other lifestyle component had a significantly reduced risk of adverse health outcomes [4]. Another systematic review of nine randomised controlled trials confirmed that a low glycaemic index (GI) diet is the most appropriate dietary intervention for women with GDM [5].

Recommendations to achieve BG targets include: (i) encouraging a healthy diet and avoiding excess energy intake and (ii) replacing high GI carbohydrates with low GI alternatives [2]. In practice, women with GDM are advised on a healthy diet including the type, amount, and distribution of carbohydrate over the day to reduce glycaemic load and subsequently the postprandial BG levels, i.e., three meals a day with 30–50 g of low GI carbohydrate and two/three snacks of 10–20 g carbohydrate [6]. However, there is insufficient evidence to demonstrate the effectiveness of this daily carbohydrate distribution on glycaemic control [7].

Furthermore, there is no evidence-based consensus on the recommended frequency and distribution of meals and snacks and their macronutrient content for GDM. One interventional study found that reducing fat intake while not imposing restrictions on carbohydrate intake in GDM women still achieved glycaemic goals [8].

This study is part of a wider project examining the feasibility and acceptability of using myfood24 [9,10,11], a self-administered online 24-h dietary recall tool, in women with GDM to monitor dietary intake.

The aim of this study is to examine the frequency, distribution, and nutritional composition of meals and snacks, in particular calorie and carbohydrate intake, in women with GDM, and to explore whether these are associated with glycaemic control.

## 2. Materials and Methods

### 2.1. Setting, Participants, and Data Collection

Women diagnosed with GDM at Leeds Teaching Hospital NHS Trust (Leeds, UK) were invited to participate in this prospective observational study, at 26–28 weeks gestation. In the UK, GDM is diagnosed by risk factor screening, then a 75 g oral glucose tolerance test at 24–28 weeks gestation [2]. Those who were unable to read and understand English or who were prescribed medication for their GDM were not eligible. A total of 200 women (aged 20–43 years) consented to take part between February 2015 and February 2016. The women continued to receive their usual care, which included dietary advice from a dietitian at their clinic visit and self-monitoring of blood glucose and food intake. For this study they were additionally asked to self-record their diet on five occasions using myfood24 over the two weeks following diagnosis (see Figure 1). The sample size of 200 was determined by a power calculation to detect difference in BG control by meeting dietary recommendations for fruit and vegetables or not (results not presented here).

Routinely collected data reported in this study include: maternal age, ethnicity, parity, height, booking weight (measured in at first midwife appointment, typically before 12 weeks gestation), and BMI.

As part of the study women were requested to submit their self-recorded capillary BG measurements, which were taken four times each day, for seven days: a fasting measurement before breakfast, and then one hour postprandially for breakfast, lunch, and dinner.

myfood24 is a web-based 24-h dietary recall tool for participants to quickly and easily record diet, according to meal type—breakfast, lunch, dinner, or snack. More information and a demonstration are available via www.myfood24.org. myfood24 has been validated against interviewer administered recall and dietary biomarkers showing favourable results [12]. Participant email addresses are uploaded, which myfood24 then uses to issue invitations. Participants had over 45,000 different food and drink products to choose from, with a range of portion size options, including portion size images, standard pack sizes, and average portions [9]. Participants received an instant summary of key dietary components (energy, protein, fat, carbohydrates, fibre, and salt) for the day, upon submission of their diet record. The research midwife explained how to use myfood24 in clinic and then the participants completed the recalls at home.

### 2.2. Ethical Approval

This project obtained ethical approval (ref 14/SC/1267), following proportionate review by the NHS Health Research Authority (NRES) Committee South Central–Oxford C in September 2014.

### 2.3. Statistical Analysis

Chi-squared tests for categorical variables and independent t-tests for continuous variables were used to report differences in maternal age, booking weight and BMI, ethnicity and parity are reported by compliance groupings (A vs. B):
completion of (A) ≥1 myfood24 versus (B) nonecompletion of myfood24 and fasting BG readings [2]:
○(A) achieving fasting BG target (<5.3 mmol/L) and○(B) exceeding the target
completion of myfood24, fasting BG readings, and all postprandial BG readings [2]:
○(A) achieving BG targets through the day: 1-h postprandial BG target (<7.8 mmol/L) and fasting BG target (<5.3 mmol/L) and○(B) exceeding the target


Fasting BG and postprandial BG were calculated as a mean of the seven days fasting BG and postprandial BG readings in each participant, respectively. For each person, the distribution of mean calories and carbohydrate intake over six eating occasions—breakfast, morning snacks and drinks (midnight to noon); lunch, afternoon snacks and drinks (noon to 6 pm); dinner, evening snacks and drinks (6 pm to midnight)—were calculated as percentages of her total mean daily intake; average percentages for the six eating occasions were then calculated over the group(s).

The percentage of meals containing 30 g to 50 g of carbohydrate and the percentage of snack and drink periods consumed containing <20 g of carbohydrate were determined.

Chi^2^ tests were used to determine whether there were significant differences between groups meeting BG targets (A) and those who did not (B) in relation to frequency of meal consumption (<3; 3 per day) and frequency of snack consumption (<2; ≥2 and <3; 3 per day).

*p* values were considered significant at <0.01, to account for multiple testing.

## 3. Results

### 3.1. General Characteristics

Of the 200 women who were recruited into this study, one withdrew, leaving *n* = 199. Participant compliance to study requirements varied, with the majority of women achieving glycaemic control following each meal (Table 1).

Of the total, 76% were overweight or obese prior to pregnancy, according to booking weight (*n* = 149/196). Significant differences exist between women who completed myfood24 (*n* = 120) and those who did not (*n* = 79) in relation to ethnicity, but not parity, age, booking weight and BMI. myfood24 completers were more likely to be white (66% vs. 44%) than non-completers (Table 2). myfood24 completers had significantly lower mean (sd) fasting BG (4.9 (0.6) mmol/L vs. 5.2 (0.5) mmol/L: *p* = 0.008) than non-completers, but not significantly lower post-breakfast, lunch or evening meal BG recordings.

Of the 102 women, 81% completed myfood24 on three or more diary days, the desirable compliance level. In order to improve the power of our analysis, we included all women who only completed one day (11.8%) and two days (7.8%) of myfood24, in addition to those completing three or more days. A total 48% of women completed five days of myfood24, the ideal level. There was no association observed between the number of days completing myfood24 and glycaemic control.

BG data was available for 157 women. Of these, 115 women recorded their fasting BG; 102 postprandial meal BG; and myfood24 on ≥1 day. Of these, women who exceeded target mean fasting BG (≥5.3 mmol/L) were significantly more likely to have a higher booking weight (mean (sd) = 90 (18) kg vs. 74 (16) kg) and BMI (34 (6) kg/m^2^ vs. 28 (6) kg/m^2^). There were no significant differences in relation to age, ethnicity, parity, or number of days to complete myfood24. Women achieving the fasting and postprandial BG targets had significantly lower mean post-breakfast, post-lunch and post-evening meal BG recordings than those not achieving the BG targets (Table 2).

### 3.2. Meal Regularity and Glucose Control

The majority (89%) of women who completed myfood24 consumed three meals a day (Table 3). A higher percentage of those achieving the fasting BG target consumed three meals a day compared to those exceeding it (92% vs. 78%: *p* = 0.04). Only 10% of women snacked in all three snacking periods during the 24-h period. Those achieving the glycaemic control throughout the day were significantly more likely to snack regularly (*p* = 0.008).

The largest percentage of energy from carbohydrate intake was consumed at evening meals (33.5% for those meeting the fasting BG target and 30.3% for those who did not). A fifth of energy was consumed at breakfast, and a fifth was consumed as snacks throughout the day. Although there were some differences in evening snack and drink intake between those meeting fasting BG targets and those not (6% vs. 4% energy, *p* = 0.03, and 7% and 3% carbohydrates, *p* = 0.01) their combined total energy (39.4% vs. 39.8%) and carbohydrate (36.8% vs. 36.6%) from evening meals and snacks were similar.

Most women consumed meals with a carbohydrate content outside the suggested 30–50 g. A total 31% of meals met this, 33% fell below, and 34% above. Only 21% of evening meals were within this range. The majority of women consumed snacks and drinks containing carbohydrates that were less than 20 g during the three snacking periods.

Women meeting the fasting BG target had higher intakes of fibre and sugar. No significant association existed with all macronutrients or fruit and vegetables for women achieving glycaemic control throughout the day (Table 4).

## 4. Discussions

We believe this is the first study to explore distribution of meals/snacks and nutrients across the day in relation to BG levels in free living western women with GDM, with meal event detail, resulting from use of a new online dietary assessment tool, myfood24 [9]. Achieving glucose targets was associated with having more snacks across the day. Other studies in GDM either reported dietary intake over the whole day or provided women with meals containing specific nutrient content [13].

### 4.1. Supporting Glycaemic Control

Maintaining good glycaemic control is a priority in the treatment of GDM, aiming to avoid peaks and troughs in BG levels. After meals BG levels are associated with the amount and type of carbohydrate and other macronutrients in the meal, with clear differences between higher GI and lower GI meals [14]. Recommendations are for even spacing of carbohydrate intake across the day, with three meals and three snacks [6,15]. The results from our pilot study indicate this may be effective for achieving BG targets, although a larger study is needed to confirm this.

Self-monitoring of BG can contribute to improving glycaemic control and meeting BG targets important for reducing the risk of macrosomia [16]. However, women may not understand what action they should take using the results of BG self-monitoring alone [17]. A combination of BG monitoring linked to dietary self-monitoring using easily accessible tools that provide immediate feedback on macronutrient intake (such as myfood24) may support behaviour change and result in better BG control. Additionally, clinicians may be able to better advise patients from the dietary feedback.

### 4.2. Diet in GDM

A systematic review of dietary advice to support management of GDM found 10 different types of diet used in randomised controlled trials (RCTs) from 19 studies. These included low-GI diets; energy restricted diets; diets rich in fruit, vegetables and whole grains, low vs. high carbohydrate diet; high unsaturated vs. low unsaturated fat diet; soy protein enriched; high fibre; behavioural advice. The studies were generally small with varying interventions and outcomes assessed. The conclusion was that evidence is still not available to guide practice [18]. A recent systematic review of 18 RCTs found that the quality of studies was generally low to very low [19]. Better measures of dietary behaviour in GDM are needed to support generation of appropriate evidence. We believe our pilot study demonstrated an appropriate tool for this research.

Another review of five RCTs showed that low-GI diets in women with GDM reduced the risk of macrosomia, which was further reduced by the addition of dietary fibre [20]. In our study, total carbohydrate intake alone was not associated with good BG control. A measure of GI was not available, however, we did find that those who met the fasting BG target had a higher fibre intake, which can be linked with lower GI. Another explanation is those exceeding fasting BG targets may under-report their food consumption. These women, exceeding the fasting BG target, had a significantly higher booking BMI and it is known that those with higher BMI are more prone to under-report their diet [21]. Difference by BMI and possible under-reporting could be investigated further in future research in a larger study.

Diet or exercise, or both, can support the risk of excessive weight gain during pregnancy [22]. As expected in this study the majority of women were overweight or obese [23]. Dietary self-monitoring is consistently associated with supporting weight loss [24]. A randomised controlled trial of a smartphone app to self-monitor diet compared with other forms of self-monitoring found the duration and frequency of app use was associated with improved weight loss [25]. Use of new technology to measure diet for women with GDM could prevent disproportionate weight gain. Antenatal dietary interventions in overweight or obese pregnant women are associated with less weight gain without a negative effect on birthweight [26,27].

The prevalence of GDM is increasing worldwide, linked to increasing rates of T2D and obesity. Strategies for prevention of GDM as well as support for treatment are needed, for the long-term health of both the mother and child. GDM mothers are at increased risk of subsequent T2D, cardiovascular disorders, and metabolic syndrome [28]. The current evidence from available RCTs on the effects of diet and exercise interventions during pregnancy to prevent GDM are limited [29,30]. A multicentre pilot RCT from nine European countries aiming to prevent GDM showed a healthy eating intervention group had less gestational weight gain and lower fasting BG than the physical activity group. A full trial with detailed dietary data is underway [31].

### 4.3. Study Strengths and Weaknesses

This study has been able to detail food and nutrient intakes across the day by meal event in women with gestational diabetes. Only one previous study from Korea has presented a similar breakdown of diet [32], until very recently when an Irish study added evidence in this area [33]. Overall energy intakes were low at 1449 kcal. In comparison, the National Diet and Nutrition Survey datum for women aged 19 to 64 years was 1595 kcal [34]. However, in a similar study of overweight and obese women with GDM, intake was higher at ~1870 kcal [33]. This may in part be due to the Ainscough et al. [33] study excluding under-reporters, where our study did not. This lower intake could reflect incomplete food recall for some days or lower intakes, perhaps as a result of pregnancy related appetite changes or their recent diagnosis and its risk to the developing baby. In this pilot study some women recorded their diet on paper and then entered all days into the system at the same time, making the dietary recall dates difficult to identify and not possible to link to day of BG measurement; therefore, procedures should be improved in a larger study. A further limitation was that the study could be under-powered to detect differences in nutrients or meal/snack events in relation to BG. The study was powered for exploration of fruit and vegetable intakes linked to 1-h postprandial BG levels, requiring 200 women. Only 102 of the 200 women recruited provided both dietary recall and all BG data. A larger study is required to generate robust results on the association between dietary components and blood glucose control, whilst controlling for confounders such as age, treatment, previous GDM, and BMI. In addition to the novel analyses currently achievable using data collected by myfood24, on-screen feedback to users and clinicians from myfood24 detailing the distribution of meals, snacks, and nutrients through the day may be helpful and could be incorporated into the tool.

## 5. Conclusions

This pilot study showed that achieving glucose targets was associated with having more snacks across the day. This aligns with current clinical practice and generates an appetite for a larger future study to investigate dietary associations with BG control in women with gestational diabetes. myfood24 is a useful tool for recording diet in this population by meal event to explore frequency and distribution of meals and snacks, rather than simply measuring just total daily intake. Illustrating how food and nutrient intakes over the day relate to glycaemic control may help provide more practical advice to women with GDM, potentially leading to better outcomes for both mother and baby.

## Figures and Tables

**Figure 1 nutrients-12-00003-f001:**
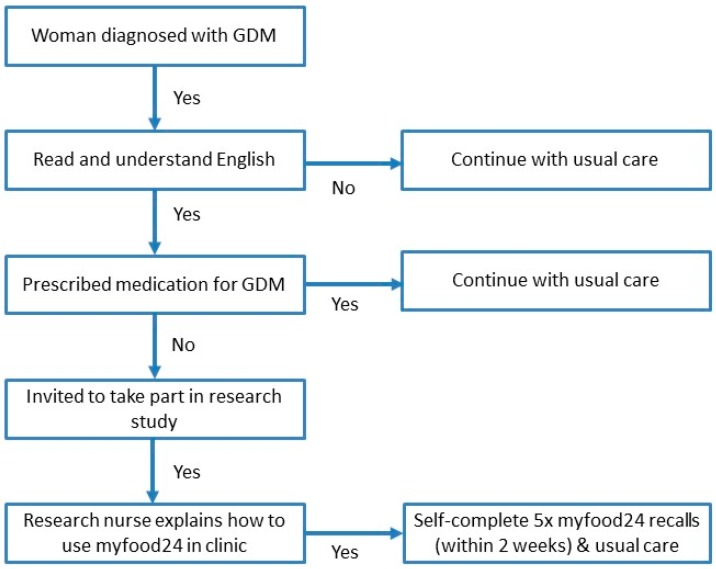
Recruitment flow chart. GDM: Gestational diabetes mellitus.

**Table 1 nutrients-12-00003-t001:** Participant compliance and achievement of Blood Glucose (BG) targets.

Study Participants	*N*
All participants	199
myfood24	
Completed	120
Did not complete	79
Monitored fasting BG and completed myfood24	
Achieved fasting BG target (<5.3 mmol/L)	88
Not achieved fasting BG target	27
Monitored postprandial breakfast BG and completed myfood24	
Achieved BG target (<7.8 mmol/L)	90
Not achieved BG target	23
Monitored postprandial lunch BG and completed myfood24	
Achieved BG target (<7.8 mmol/L)	106
Not achieved BG target	9
Monitored postprandial evening meal BG and completed myfood24	
Achieved BG target (<7.8 mmol/L)	100
Not achieved BG target	15
Monitored all postprandial meals and completed myfood24	
Achieved BG target (<7.8 mmol/L)	80
Not achieved BG target	33
Monitored fasting BG and all postprandial meals and completed myfood24	
Achieved all BG targets	68
Not achieved all BG targets	34

**Table 2 nutrients-12-00003-t002:** General characteristics of GDM participants.

		myfood24			Monitored Fasting BG and Completed myfood24			Monitored Fasting BG, All Postprandial Meals BG and Completed myfood24		
	All (*n* = 199)	Completed (*n* = 120)	Did Not Complete (*n* = 79)	*p* Value	Achieved Fasting BG Target (*n* = 88)	Not Achieved Fasting BG Target (*n* = 27)	*p* Value	Achieved BG Target (*n* = 68)	Not Achieved BG Target (*n* = 34)	*p* Value
Age, years: mean (sd)	33.3 (5.0)	33.4 (4.6)	33.2 (5.6)	0.8	33.1 (4.6))	34.2 (4.4)	0.3	32.9 (4.4)	34.3 (4.9)	0.2
Booking weight, kg (sd)	78.4 (18.4) ^a^	77.8 (17.8)	79.1 (19.3)	0.7	73.9 (16.4)	89.5 (17.9)	0.001	75.9 (16.8)	76.9 (19.3)	0.8
Booking BMI, kg/m^2^ (sd)	29.7 (6.5) ^a^	29.4 (6.2)	30.1 (7.0)	0.5	28.0 (5.5)	33.8 (6.1)	<0.001	28.4 (5.8)	29.8 (6.8)	0.3
Ethnicity *n* (%)				<0.001			0.3			0.05
White	114 (57.3%)	79 (65.8%)	35 (44.3%)		60 (68.2%)	16 (59.3%)		51 (75.0%)	17 (50.0%)	
Asian	54 (27.1%)	31 (25.8%)	23 (29.1%)		19 (21.6%)	10 (37.0%)		11 (16.2%)	14 (41.2%)	
Black	20 (10.1%)	3 (2.5%)	17 (21.5%)		3 (3.4%)	0 (0.0%)		2 (2.9%)	1 (2.9%)	
Mixed, Chinese, Other	11 (5.5%)	7 (5.8%)	4 (5.1%)		6 (6.8%)	1 (3.7%)		4 (5.9%)	2 (5.9%)	
Parity *n* (%)				0.02			0.4			0.3
Primiparous	73 (36.7%)	52 (43.3%)	21 (26.6%)		39 (44.3%)	10 (37.0%)		32 (47.1%)	12 (35.3%)	
Multiparous	126 (63.3%)	68 (56.7%)	58 (73.4%)		49 (55.7%)	17 (63.0%)		36 (52.9%)	22 (64.7%)	
Days completed myfood24 (sd)		3.8 (1.4)	-		3.8 (1.4)	4.0 (1.3)	0.5	3.8 (1.5)	3.8 (1.3)	1.0
Mean fasting BG mmol/L (sd)	5.0 (0.7) ^b^	4.9 (0.6)	5.2 (0.5)	0.008	4.9 (0.6)	5.8 (0.5)	<0.001	4.7 (0.4)	5.2 (0.8)	<0.001
Mean post breakfast BG mmol/L (sd)	7.2 (1.0) ^b^	7.1 (1.0)	7.2 (1.1)	0.6	6.9 (0.8)	7.9 (1.2)	<0.001	6.6 (0.7)	8.1 (1.0)	<0.001
Mean post lunch BG mmol/L (sd)	6.7 (1.0) ^b^	6.6 (0.9)	7.0 (1.0)	0.02	6.5 (0.7)	7.0 (1.3)	0.008	6.4 (0.6)	7.2 (1.2)	<0.001
Mean post dinner BG mmol/L (sd)	6.8 (1.0) ^b^	6.7 (1.0)	7.1 (1.1)	0.03	6.5 (0.7)	7.3 (1.3)	<0.001	6.4 (0.6)	7.6 (1.2)	<0.001

^a^*n* = 196 booking weight and BMI. ^b^
*n* = 176 to 178 (about 95% of those who completed myfood24, and 80% of those who did not, provided their BG readings). *p* values determined using t-test and chi^2^ tests.

**Table 3 nutrients-12-00003-t003:** Meal regularity of GDM participants.

		Monitored Fasting BG and Completed myfood24			Monitored Fasting BG and All Postprandial Meals BG and Completed myfood24		
	All myfood24 (*n* = 120)	Achieved (*n* = 88)	Not Achieved (*n* = 34)	*p* Value	Achieved (*n* = 68)	Not Achieved (*n* = 34)	*p* Value
Frequency of meal consumption *n* (%)				0.04			0.05
Fewer than three times a day	13 (10.8%)	7 (8.0%)	6 (22.2%)		7 (10.3%)	0 (0.0%)	
Three times a day	107 (89.2%)	81 (92.0%)	21 (77.8%)		61 (89.7%)	34 (100%)	
Frequency of snack and drink consumption *n* (%)				0.06			0.008
Fewer than two snacking periods a day	62 (51.7%)	39 (44.3%)	19 (70.4%)		26 (38.2)	24 (70.6)	
≥twice and < three snacking periods a day	48 (40.0%)	40 (45.5%)	7 (25.9%)		35 (51.5)	8 (23.5)	
Three snacking periods a day	10 (8.3%)	9 (10.2%)	1 (3.7%)		7 (10.3)	2 (5.9)	
Distribution of mean energy intake % (sd)							
Breakfast	19.6 (7.1)	19.1 (6.4)	21.1 (9.5)	0.5	19.1 (6.3)	20.1 (6.9)	0.5
Morning snack and drinks	6.2 (7.0)	6.3 (6.4)	6.5 (9.0)	0.4	6.0 (6.3)	6.7 (8.0)	0.9
Lunch	28.0 (9.7)	28.1 (9.4)	27.2 (11.4)	0.8	28.3 (9.5)	29.3 (10.0)	0.7
Afternoon snack and drinks	6.5 (5.9)	6.9 (5.8)	5.6 (6.0)	0.2	6.9 (5.9)	6.3 (4.9)	0.8
Dinner	34.4 (9.0)	33.4 (8.4)	36.1 (10.1)	0.2	33.7 (8.8)	33.2 (8.0)	1.0
Evening snack and drinks	5.3 (6.0)	6.0 (6.2)	3.5 (5.2)	0.03	6.0 (6.6)	4.4 (4.5)	0.5
Distribution of mean carbohydrate intake % (sd)							
Breakfast	20.9 (8.6)	20.4 (7.3)	21.6 (11.7)	0.9	20.4 (6.6)	21.2 (10.5)	0.9
Morning snack and drinks	7.8 (9.0)	7.6 (7.9)	8.7 (12.4)	0.7	7.3 (7.3)	8.4 (11.6)	0.7
Lunch	26.8 (9.5)	27.1 (9.0)	26.2 (11.9)	0.7	26.7 (9.3)	29.3 (9.1)	0.3
Afternoon snack and drinks	7.6 (6.5)	8.02 (6.6)	6.9 (6.3)	0.3	8.0 (6.6)	7.4 (5.5)	0.8
Dinner	31.4 (11.4)	30.3 (11.0)	33.5 (12.2)	0.1	30.9 (12.0)	29.6 (9.1)	0.9
Evening snack and drinks	5.6 (7.1)	6.5 (7.5)	3.3 (5.5)	0.01	6.7 (8.1)	4.0 (4.3)	0.2
Percentage (sd) meals between 30 and 50 g							
carbohydrate at	41.8 (34.4)	42.6 (34.4)	41.4 (37.2)	0.8	43.2 (34.3)	38.7 (37.0)	0.5
Breakfast	31.5 (28.0)	30.6 (27.6)	35.2 (30.4)	0.5	27.4 (26.0)	37.9 (28.8)	0.08
Lunch	20.7 (25.8)	20.3 (24.1)	23.8 (32.4)	1.0	19.5 (22.6)	20.5 (25.2)	1.0
Evening meal							
Percentage (sd) snack and drink periods >0 to ≤20 g carbohydrate							
Morning	73.0 (32.6)	70.8 (32.8)	78.4 (32.7)	0.2	70.6 (33.6)	76.6 (31.0)	0.4
Afternoon	73.1 (31.1)	70.9 (32.8)	76.9 (26.7)	0.5	70.9 (33.0)	72.9 (29.6)	1.0
Evening	81.5 (25.7)	78.1 (27.4)	90.6 (17.7)	0.02	77.2 (29.4)	86.4 (18.2)	0.2

Drinks that include kcals; Snacking periods are: morning (midnight to before noon); afternoon (noon to before 6 pm); evening (6 pm to before midnight); *p* values were determined using chi^2^ tests relating to frequency of meal and snack consumption and using Mann–Whitney (Wilcoxon rank–sum) tests for the other associations since the majority of variables had skewed distributions.

**Table 4 nutrients-12-00003-t004:** Mean daily macronutrient and fruit and vegetable intake.

		Monitored Fasting BG and Completed myfood24			Monitored Fasting BG and All Postprandial Meals and Completed myfood24		
	All myfood24 (*n* = 120)	Achieved (*n* = 88)	Not Achieved (*n* = 27)	*p* Value	Achieved (*n* = 68)	Not Achieved (*n* = 34)	*p* Value
Energy, kcal mean (sd)	1449 (389)	1485 (385)	1343 (421)	0.2	1468.7 (1382.6)	1515.6 (439.9)	0.9
Carbohydrate, g (sd)	170 (49.4)	174 (50)	156 (46)	0.1	172.6 (44.1)	177.3 (59.0)	0.9
Protein, g (sd)	63.2 (18.6)	64.1 (16.9)	61.0 (23.4)	0.4	62.2 (16.3)	69.5 (19.0)	0.1
Fat, g (sd)	58.7 (21.6)	60.2 (21.4)	53.1 (22.4)	0.1	59.8 (21.7)	59.4 (19.8)	0.9
Dietary fibre, g (sd)	16.2 (6.0)	17.0 (6.1)	13.7 (5.6)	0.01	16.4 (5.8)	17.0 (6.5)	0.8
Sugar, g (sd)	53.9 (26.1)	56.4(23.5)	48.0 (33.2)	0.01	58.9 (24.2)	50.5 (28.3)	0.03
Saturated fat, g (sd)	19.9 (8.4)	20.4 (8.6)	18.3 (8.1)	0.2	20.4 (9.1)	19.4 (6.8)	0.7
Sodium g (sd)	2.2 (0.9)	2.2 (0.8)	2.2 (0.8)	0.8	2.1 (0.7)	2.5 (1.1)	0.4
% (sd) Energy from Carbohydrates	46.9 (8.0)	47.4 (7.5)	47.5 (8.7)	0.8	47.6 (7.7)	46.9 (6.8)	0.5
% (sd) Energy from Fats	35.4 (6.9)	35.9 (6.8)	35.0 (6.7)	0.5	35.9 (7.0)	35.2 (5.8)	0.6
% (sd) Energy from Proteins	17.9 (3.5)	17.5 (3.3)	18.1 (3.9)	0.3	17.2 (3.3)	18.6 (3.0)	0.02
Total vegetables g (sd)	149 (98)	156 (102)	121 (82)	0.08	156.1 (108.5)	142.4 (89.0)	0.6
Total fruit g (sd)	140 (101	145 (100)	132 (103)	0.4	146.8 (103.9)	136.2 (91.1)	0.7
Total fruit and vegetables g (sd)	274 (147)	285 (144)	234 (160)	0.06	285.3 (149.7)	262.6 (145.2)	0.4

*p* values were determined using Mann–Whitney (Wilcoxon rank–sum) tests for non-parametric distributions.

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
