# Peer review of "Relationship of the Frequency, Distribution, and Content of Meals/Snacks to Glycaemic Control in Gestational Diabetes: The myfood24 GDM Pilot Study"

_nutrients, 2019, doi:10.3390/nu12010003_

Round 1

Reviewer 1 Report

Fantastic study and well-written manuscript, was a privilege to review.

Just a few minor revisions:

1) line 19/20 change order of sentences. "Diet was assessed using myfood24, a novel..... 115/200 women completed ..."
2) change in line 19 "both" to "115/200 women completed ≥1 dietary recall and recorded their blood glucose measurements."
3) numbers don't seem to match with table numbers - shouldn't 115 be 102?
4) remove comma in line 18
5) remove commas in line 22
6) please comment on how you identified women's breakfast, lunch, and dinner serves - are these specified in the myfood24?
7) why were snacks/drinks with <20g of carbohydrate considered and not <30g?
8) check tables for appropriate bolding of headers - inconsistent in the version I received
9) when table runs over more than 1 page, please include headers on all pages (easier for reader to follow)
10) table 2 columns under "monitored fasting bg and completed myfood24" when referring to achieved bg target, is this just for the fasting target? Please specify in table.
11) unclear in table 3 what the numbers represent in frequency of meal consumption and frequency of snack and drink consumption rows (assume n (%) but please specify)
12) numbers in bottom right of table 3 on page 7 have no units, please correct and check consistency throughout table.
13) is % (sd) in table 3 mean (sd) and unit is %? If so, please correct how this is written. However, some of the data is skewed (as mentioned in the table footnote + clear in the size of some of the sds compared to means (if they are means presented)) so these should be presented as median (lower quartile, upper quartile)
14) lines 209-224 aren’t relevant to the study conducted or findings from the study, suggest removal from the manuscript.
15) line 228 please correct “in comparison the”
16) line 250 please add comma after GDM

Reviewer 2 Report

General comment:

This study provides detailed food and nutrient intakes across the day by meal events among a sample of women with gestational diabetes. The findings might help to frame practical advice on meal and snack patterns and frequency during pregnancy. Regardless the limitations, this study adds valuable evidence to this field of research. The manuscript is worthy of publication provided the following matters are addressed.

Title and the abstract:

The title accurately reflects the content and, in general, the abstract presents an adequate synopsis of the paper.

Introduction:

The introduction provides a good, generalized background of the topic with logically organized, clear and well-argued narrative. The objective is clearly defined.

Materials and methods:

Please provide information on time frame of dietary data collection and time-wise congruence of glucose blood level analyses and 24h dietary recalls. As it is stated in the article, participants were asked to self-record their diet on five occasions using myfood24 over the two weeks following diagnosis. On the other hand, they were supposed to submit their self-recorded capillary blood glucose measurements taken four times each day, for seven days. Were these periods matched? Were there any rules with regards to distribution between workdays and weekend days?  Was diet recorded on consecutive or non-consecutive days?

Major issue: Please provide the rationale for considering participants who completed at least one 24h recall (i.e. 20% of the anticipated number of recalls) via myfood24 compliant? Among 24h dietary recall completers how many achieved ideal compliance level (i.e. recorded diet for 5 days)?

Please provide the rationale for recruiting 200 women.  Was the number pre-defined due to study power calculation, or that is the number of females who fulfilled the inclusion criteria during the one-year period of study duration?

Results/Discussion:

In Table 2, in rows referring to mean fasting/post-breakfast/post-lunch/post-dinner blood glucose level,  please indicate what does the number in the brackets refer to (probably sd).

Was the increased number of provided 24h recalls associated with improved glycaemic control?

Line210: “As expected in this study the majority of women were overweight or obese.” Please, provide a reference to support this notion.

Given that the main aim of the study was to examine the frequency, distribution and nutritional composition of meals and snacks, in particular calorie and carbohydrate intake, it might be useful to provide dietary pattern analysis with regards to food group distribution.

As it is stated in the Limitations section of the article, overall energy intakes were low at 1449kcal. This finding was compared against the National Diet and Nutrition Survey data for women aged 19 to 64 years. However, no data (derived from other studies) is presented for estimated energy intake for pregnant women (especially those with diagnosed gestational diabetes).  Please, indicate whether the assessment of energy under-reporting was performed. If yes, elaborate on the method applied.

For additional discussion, please consider the following recently published articles:

Ainscough KM, Kennelly MA, Lindsay KL, O’Brien EC, O’Sullivan EJ, Mehegan J, Gibney ER, McAuliffe FM. An observational analysis of meal patterns in overweight and obese pregnancy: exploring meal pattern behaviours and the association with maternal and fotal health measures. Irish Journal of Medical Science (1971-). 2019 Nov 15:1-0.

Petry CJ, Ong KK, Hughes IA, Acerini CL, Dunger DB. Temporal Trends in Maternal Food Intake Frequencies and Associations with Gestational Diabetes: The Cambridge Baby Growth Study. Nutrients. 2019 Nov;11(11):2822.
